# Transitioning from Full-Context to Active Evidence-Seeking Evaluation: A Novel Benchmark for Real-World Artificial Intelligence Assisted Medical Diagnosis

## Abstract

Large language models (LLMs) achieve strong results on medical benchmarks, yet prevailing evaluations rely on a passive, full-context paradigm (where complete information is provided upfront), failing to reflect clinical practice where information is scarce, cues are ambiguous, and clinicians must proactively elicit and verify evidence. Such static designs bypass the most critical stage, active evidence-seeking, and systematically overestimate model capability. We introduce **ROUNDS-Bench**, which decouples information-acquisition strategy from diagnostic reasoning and uses a standardized patient simulator to reconstruct multi-turn active evidence-seeking diagnostic processes (history-taking, physical exam, test ordering). The benchmark comprises two tasks: **Task 1 (Full-Context)** provides complete cases to estimate performance upper bounds; **Task 2 (Active Evidence-Seeking)** reveals only demographics and chief complaint, requiring models to proactively drive multi-turn questioning/test selection, stop evidence-gathering at appropriate points, and deliver diagnoses. Evaluations of state-of-the-art LLMs (e.g., GPT-4o, Qwen, DeepSeek, Llama) show substantial degradation from Task 1 to Task 2, exposing a capability gap between passive evaluation and real clinical decision-making and highlighting the need for improved active evidence-seeking and decision integration. ROUNDS-Bench aims to shift medical AI from passive answering toward proactive agents that inquire, investigate, halt timely, and diagnose accurately advancing reliable, efficient, and safe clinical decision support. We will release code and simulator interfaces for reproducibility.

## 1 Introduction

Medical AI evaluation currently faces a key limitation: **while large language models (LLMs) achieve impressive results on static medical benchmarks (Liao et al., 2024), demonstrating tangible potential for clinical decision support, prevailing evaluations predominantly rely on a passive, full-context paradigm (single-shot provision of complete case histories) (Liu et al., 2025a). This is fundamentally misaligned with clinical reality.** Diagnosis is a dynamic process that functions under incomplete information and constant uncertainty(Coderre et al., 2003). Clinicians must actively guide history-taking, physical examination, and test ordering, and they must determine appropriate points at which to stop gathering evidence to reach diagnostic conclusions (Li et al., 2024). Bypassing this core active evidence-seeking stage risks systematically overestimating model capability and obscuring critical weaknesses in active evidence-seeking strategy and clinical safety (e.g., misdiagnosis risks). *How, then, can we scientifically, reproducibly, and clinically evaluate this active capability?* Mature practices from medical education offer key insights.

In medical education, written examinations assess knowledge acquisition statically but fail to simulate dynamic clinical information flow and resource-risk constraints. Consequently, assessing dynamic diagnostic competence in real-world contexts typically employs the Objective Structured Clinical Examination (OSCE) (Khan et al., 2013). OSCE leverages Standardized Patients (SPs) to create controlled, reproducible scenarios covering key stages like inquiry, preliminary assess-

ment, test selection, and decision-making(Barrows, 1993). It evaluates not only the correctness of diagnostic conclusions but also the rationality of the evidence path (i.e., the strategy and process for actively acquiring information) and the quality of the evidence chain (relevance, sufficiency, consistency). This paradigm indicates that clinical artificial intelligence evaluation must transcend "passive answering" to assess abilities to proactively initiate inquiry, rationally plan examinations, and judiciously decide when to stop, requiring the evaluated agent to make coherent strategic choices about what evidence to obtain and when(Yao et al., 2024)Jiang et al. (2025).

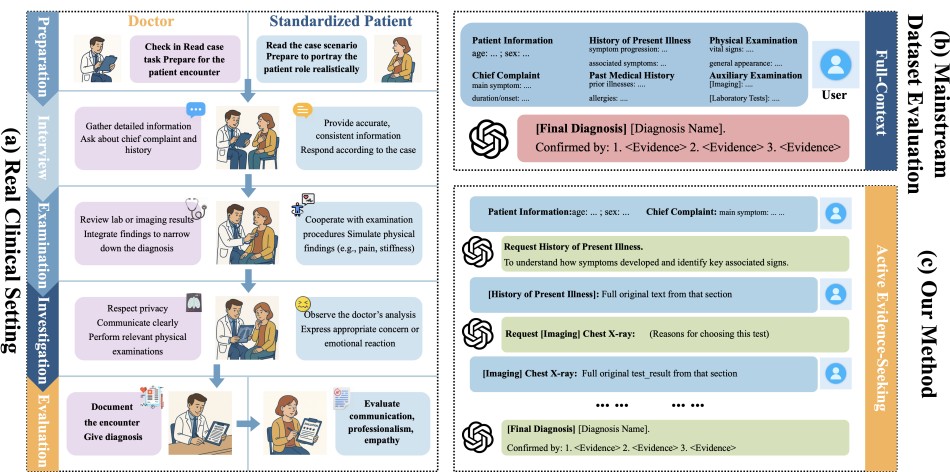

Figure 1: Evaluation paradigms for clinical diagnostic reasoning. **(a) Real Clinical Setting**: Standardized Patients (SPs) simulate full clinical encounters in OSCE. **(b) Mainstream Dataset Evaluation**: The model is given full patient information at once and directly produces a diagnosis in a single turn. **(c) Our Method**: The model interacts in a multi-turn manner by actively requesting specific categories of information, and synthesizing the evidence to determine when to diagnose.

Therefore, we introduce Rounds-based Ongoing Understanding for N-turn Diagnosis (**ROUNDS-Bench**), constructing an interactive diagnostic evaluation benchmark rigorously aligned with the core spirit of OSCE. Its essence lies in decoupling "active evidence-seeking strategy" from "diagnostic reasoning". We design a standardized Patient Simulator (SP-sim) that returns information only upon explicit requests from the model (acting as the clinician), such as specific history details, physical exam findings, or lab/imaging results—providing no unsolicited clues. This ensures interaction controllability, reproducibility, and sensitivity to strategic differences. To reconcile ceiling and process-aligned evaluation perspectives, ROUNDS-Bench comprises two complementary tasks: Task 1 (Full-Context) provides complete cases to characterize the inference upper bound under information sufficiency; Task 2 (Active Evidence-Seeking) reveals only demographics and chief complaint, requiring the model to proactively drive multi-turn evidence request sequences (spanning history, exam, tests) and autonomously decide when to stop evidence gathering and deliver a final diagnosis.

Unlike frameworks emphasizing multi-objective process metrics, this work converges evaluation onto two endpoints: (i) final diagnostic accuracy and (ii) evidence quality. Here, "evidence quality" assesses whether the evidence set relied upon at diagnosis satisfies:(1) High diagnostic value (relevance) and sufficiency; (2) Consistency and traceability in the evidence-to-conclusion chain; (3) Necessary coverage of key discriminative findings.Scoring follows predefined criteria (see Methods), emphasizing "using the right information sufficiently" over redundant accumulation, thereby distinguishing "arriving at the correct diagnosis" from "arriving at the correct diagnosis supported by high-quality evidence".

Using this framework, we systematically evaluate advanced LLMs from leading families (e.g., GPT-4o, Llama (Grattafiori et al., 2024), Qwen (Yang et al., 2025), DeepSeek (Guo et al., 2025)). Empirical results show substantial performance degradation in Task 2 (requiring active evidence-seeking) compared to Task 1, with shared failure modes: Ineffective/redundant evidence gathering (requests with low diagnostic yield or marginal information gain); Premature diagnostic closure with insufficient evidence (concluding before acquiring critical cues); Test request sequence deviating from

clinical pathways/constraints (ignoring cost, contraindications, or typical workflows). These findings reveal significant gaps in driving active evidence-seeking strategies and effectively integrating evidence, highlighting the imperative for strategy alignment and training tailored to real clinical workflows.

Our main contributions are summarized as follows:

**1.ROUNDS-Bench**:A clinically aligned, interactive active evidence-seeking evaluation benchmark centered on a request–response SP-sim, systematically characterizing abilities to "proactively ask, investigate, stop timely, and diagnose accurately".

**2.Reproducible experimental protocol, code & interfaces**: Explicit specification of request granularity, visibility scope, and return protocols, ensuring interaction controllability and experimental reproducibility.

**3.Two-endpoint metric establishment**: Final diagnostic accuracy and evidence quality, respectively measuring diagnostic conclusion correctness and the rationality of the supporting evidence.

**4.Empirical revelation of the capability gap**: Validation of benchmark sensitivity across advanced LLMs, uncovering systemic challenges and characteristic failure modes in interactive evidence-seeking scenarios, providing clear targets and reusable baselines for future methods.

## 2 RELATED WORK

### 2.1 STATIC FULL-CONTEXT MEDICAL BENCHMARKS.

Early benchmarks in medical NLP and clinical QA primarily adopt a static format: the model receives a complete question and limited context in a single input to produce an answer (Roberts et al., 2020). Representative datasets include MedQA-USMLE (Jin et al., 2021),, which leverages U.S. medical licensing exam questions to assess multi-disciplinary reasoning via multiple-choice formats; PubMedQA (Jin et al., 2019), which centers on biomedical research questions and abstracts, requiring binary or uncertain judgments grounded in evidence; and MedMCQA (Pal et al., 2022), a large-scale dataset of nearly 200,000 Indian medical exam questions covering 21 subjects and over 2400 concepts. More recently, MultiMedQA (Singhal et al., 2023) unified various QA forms (exam-style, research, and consumer health) and led to systems like Med-PaLM2 (Singhal et al., 2025), which apply instruction tuning and chain-of-thought (Wei et al., 2022) prompting to enhance clinical understanding. While these efforts significantly advanced large-scale comparison of medical knowledge coverage and answer accuracy, they exhibit key limitations: information is fully exposed at once, entangling retrieval with reasoning; and they lack evaluation of evidence chains or diagnosis credibility (Markus et al., 2021).

### 2.2 ACTIVE EVIDENCE-SEEKING & CLINICAL SIMULATION.

Benchmarks aligned with real clinical workflows draw inspiration from medical education, particularly OSCEs (Khan et al., 2013). Harden and Gleeson introduced structured multi-station assessments of diagnostic ability (Harden & Gleeson, 1979), while Barrows pioneered the use of SPs to simulate clinical encounters (Barrows, 1993). In recent years, data resources have emerged to support dialog-driven modeling, such as domain-specific consultations (e.g., for respiratory complaints), bilingual corpora like MedDialogChen et al. (2020), and structured history-taking datasets like MediTOD(Saley et al., 2024). These resources facilitate modeling of "ask–examine–diagnose" workflows, but often suffer from passive information disclosure or coarse interaction granularity, making it hard to simulate request-driven evidence collection. Methodologically, researchers have modeled diagnosis as a sequential decision process , applying POMDPs or active feature acquisition frameworks to learn when to stop gathering information under cost constraints (Tsoukalas et al., 2015; von Kleist et al., 2025). These approaches support more faithful simulations of clinical reasoning and the optimization of evidence-seeking strategies .

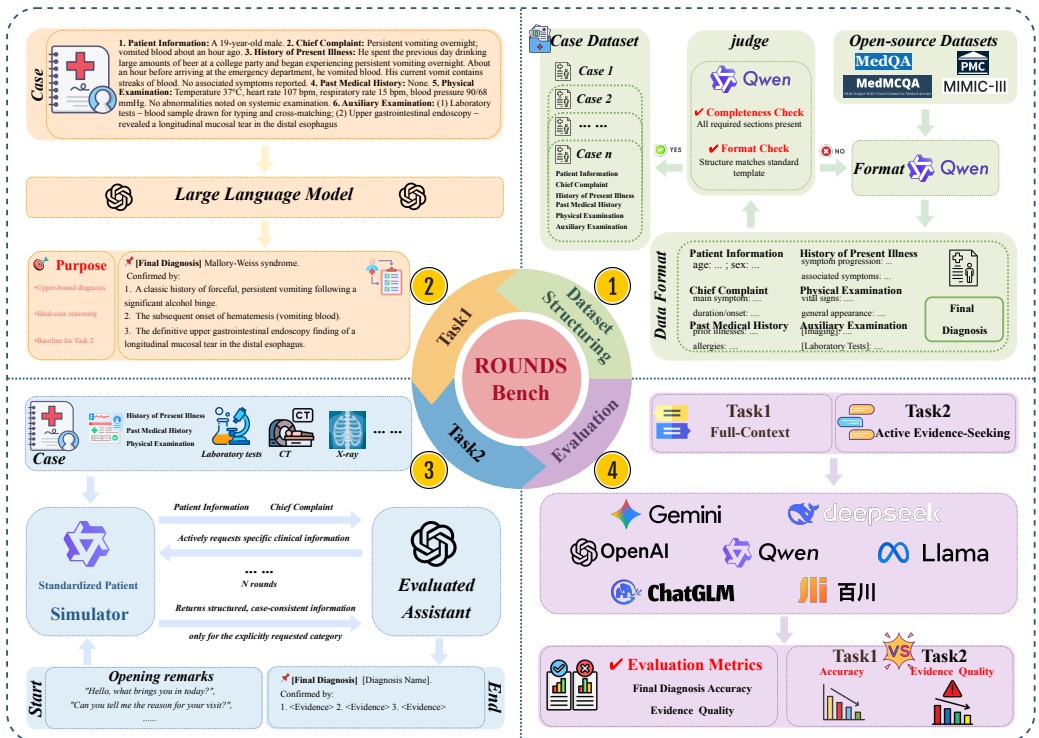

Figure 2: **Overview of ROUNDS-Bench.** The benchmark consists of four modules: (1) **Dataset Structuring** standardizes multi-section clinical cases; (2) **Task 1** evaluates models under full-context input to estimate diagnostic upper bounds; (3) **Task 2** requires models to actively seek evidence through multi-turn interaction with a patient simulator; (4) **Evaluation** assesses models based on final diagnosis accuracy and evidence quality. Results across major LLMs reveal performance gaps between passive and active diagnostic settings.

## 3 METHODS

ROUNDS-Bench evaluates *active evidence-seeking diagnosis* through a controlled, request–response pipeline (Fig. 2). The design pursues two complementary objectives: (i) to **decouple** information acquisition strategies from diagnostic reasoning, and (ii) to **enforce realism** via a standardized patient simulator (SP-sim), which only returns explicitly requested information.

The pipeline comprises four modules: (1) *Dataset Structuring* of multi-section clinical cases; (2) *Task 1 (Full-Context Diagnosis)*, which estimates diagnostic upper bounds under complete information; (3) *Task 2 (Active Evidence-Seeking Diagnosis)*, which requires multi-turn requests, stopping decisions, and final predictions; (4) *Evaluation*, which jointly considers diagnostic accuracy and evidence quality. We elaborate on each module below.

### 3.1 DATASET CONSTRUCTION

To construct a unified benchmark, we integrate four widely used resources: MedQA-USMLE (Jin et al., 2021), MedMCQA (Pal et al., 2022), MedFound (Liu et al., 2025b), and MedCase (Wu et al., 2025). For MedQA and MedMCQA, only *single-answer diagnostic cases* are retained; recall-style, multi-choice, or non-diagnostic items are excluded. See Appendix A for detailed.

*Design rationale.* Consolidating heterogeneous datasets ensures coverage of diverse clinical presentations while maintaining a controlled label space and uniform case structure. This is essential for fair comparison across models under identical assumptions.

Each clinical case is represented as a structured tuple:

$$\text{Case} = (\mathcal{P}, \mathcal{C}, H, PMH, PE, AE, \text{Dx}), \tag{1}$$

where $\mathcal{P}$ denotes patient demographics, $\mathcal{C}$ the chief complaint, $H$ the history of present illness (HPI), $PMH$ the past medical history, $PE$ the physical examination findings, $AE$ auxiliary examinations such as laboratory tests or imaging, and $\text{Dx}$ the gold-standard diagnosis.

All cases are normalized into a markdown-style sectioned format by a deterministic parser $\phi(\cdot)$. Gold-standard diagnoses are mapped to SNOMED-CT or ICD-10 codes to ensure ontology alignment. Synonyms are preserved to increase robustness in evaluation. Cases exhibiting logical contradictions (e.g., "male pregnancy") are removed. To ensure diversity and balance, we apply a stratified split across disease categories and annotate each case with difficulty levels based on diagnostic ambiguity and evidence sparsity.

## 3.2 TASK 1: FULL-CONTEXT DIAGNOSIS

Task 1 evaluates diagnostic reasoning under idealized conditions where all relevant clinical information is provided upfront. Formally, each model is presented with the full case:

$$\mathcal{X}_{\text{full}} = [\mathcal{P}, \mathcal{C}, H, PMH, PE, AE]. \tag{2}$$

The model generates both a predicted diagnosis $\hat{y}$ and a supporting evidence list $\hat{E}$:

$$(\hat{y}, \hat{E}) = f_\theta(\mathcal{X}_{\text{full}}), \quad \hat{E} = \{e_1, e_2, e_3\}, \tag{3}$$

where $f_\theta(\cdot)$ denotes the diagnostic reasoning model parameterized by $\theta$, and $\hat{E}$ contains exactly three items extracted from the case.

To ensure comparability, outputs follow a strict format: an explicit tag [Final Diagnosis] followed by three numbered evidence items. Each $\hat{y}$ is normalized by a concept mapping $\nu(\cdot)$ that aligns free-text outputs with ontology terms. Hyperparameters for generation (e.g., temperature, max tokens) are fixed across experiments to guarantee reproducibility.

*Rationale.* Task 1 establishes the **upper bound** of diagnostic performance, since information acquisition is not a confounding factor. This serves as a reference point for Task 2, where the challenge lies in sequential evidence gathering.

## 3.3 TASK 2: ACTIVE EVIDENCE-SEEKING DIAGNOSIS

Task 2 simulates a more realistic diagnostic workflow in which models must proactively request information. Initially, the model observes only patient demographics and the chief complaint: $s_0 = [\mathcal{P}, \mathcal{C}]$. At each interaction step $t$, the model issues a discrete action $a_t \in \mathcal{A}$, where the action space $\mathcal{A}$ is defined as:

$$\begin{aligned} \mathcal{A} = \{&\text{HPI}, \text{PMH}, \text{PE}\} \\ &\cup \{\text{LAB}(x), \text{IMG}(x), \text{FUNC}(x), \text{PANEL}(x)\} \\ &\cup \{\text{STOP\_DIAG}\}. \end{aligned} \tag{4}$$

Here, $\text{HPI}, \text{PMH}, \text{PE}$ denote requests for structured clinical sections, while $\text{LAB}(x), \text{IMG}(x), \text{FUNC}(x), \text{PANEL}(x)$ specify auxiliary investigations with argument $x$ indexing a particular test. The terminal action $\text{STOP\_DIAG}$ signals readiness to provide a diagnosis.

The SP-sim returns only the explicitly requested section:

$$o_{t+1} = \mathcal{R}(a_t, \text{Case}), \quad s_{t+1} = s_t \cup \{o_{t+1}\}, \tag{5}$$

where $\mathcal{R}$ is the retrieval function mapping an action to the corresponding section of the ground-truth case.

Upon issuing $\text{STOP\_DIAG}$, the model must produce a final prediction $(\hat{y}, \hat{E})$, where:

$$\hat{E} \subseteq \{o_1, o_2, \ldots, o_T\}, \tag{6}$$

ensuring that all cited evidence originates strictly from retrieved content.

*Constraints.* Each turn permits exactly one request, prohibiting batch queries or unsolicited information. This enforces a realistic, disciplined evidence-seeking strategy analogous to actual physician–patient encounters.

## 3.4 EVALUATION METRICS

We evaluate both the **accuracy of the final diagnosis** and the **quality of supporting evidence**. These metrics are shared across Task 1 and Task 2, ensuring comparability. Unless otherwise specified, $N$ denotes the number of cases. An external evaluator (Qwen2.5-32B-Instruct) scores all outputs using fixed prompts and deterministic decoding.

### 3.4.1 FINAL DIAGNOSIS ACCURACY

For the $i$-th case, let $\hat{y}^{(i)}$ denote the model-predicted diagnosis and $y^{*(i)} \in \mathcal{Y}$ the gold standard. A normalization function $\phi : \text{text} \to \mathcal{Y}$ maps predictions into the ontology label space. Since gold references are pre-normalized, $\phi$ acts as the identity during evaluation.

A discrete score $s_{\text{acc}}^{(i)} \in \{0, 1, 2\}$ is assigned as follows:

- 2: fully correct (exact disease and subtype match);
- 1: partially correct (correct disease family or category, but imprecise subtype/terminology);
- 0: incorrect.

We report two aggregate metrics:

$$\text{ExactAcc} = \frac{1}{N} \sum_{i=1}^{N} \mathbb{I}[s_{\text{acc}}^{(i)} = 2], \tag{7}$$

$$\text{GradedAcc}_\alpha = \frac{1}{N} \sum_{i=1}^{N} \left( \mathbb{I}[s_{\text{acc}}^{(i)} = 2] + \alpha \cdot \mathbb{I}[s_{\text{acc}}^{(i)} = 1] \right), \tag{8}$$

where $\alpha \in [0, 1]$ controls the weight assigned to partial correctness (default $\alpha = 0.5$).

### 3.4.2 EVIDENCE QUALITY (EQ)

Each model must submit exactly three supporting items: $\hat{\mathcal{E}}^{(i)} = \{e_1^{(i)}, e_2^{(i)}, e_3^{(i)}\}$. The evidence must be directly retrievable from the accessible content:

$$\hat{\mathcal{E}}^{(i)} \subseteq \begin{cases} x_{\text{full}}^{(i)}, & \text{Task 1,} \\ \mathcal{S}_T^{(i)}, & \text{Task 2,} \end{cases} \tag{9}$$

where $x_{\text{full}}^{(i)}$ is the full case record and $\mathcal{S}_T^{(i)}$ is the accumulated set of retrieved sections at termination.

A discrete score $s_{\text{eq}}^{(i)} \in \{0, 1, 2\}$ is assigned:

- 2: all three evidence items are present and strongly support the diagnosis;
- 1: partially supported (only 1–2 valid items or evidence is weakly inferable);
- 0: unsupported, fabricated, or contradictory.

We report both strict and graded metrics:

$$\text{EQ}_{\text{strict}} = \frac{1}{N} \sum_{i=1}^{N} \mathbb{I}[s_{\text{eq}}^{(i)} = 2], \tag{10}$$

$$\text{EQ}_\beta = \frac{1}{N} \sum_{i=1}^{N} \left( \mathbb{I}[s_{\text{eq}}^{(i)} = 2] + \beta \cdot \mathbb{I}[s_{\text{eq}}^{(i)} = 1] \right), \tag{11}$$

where $\beta \in [0, 1]$ (default $\beta = 0.5$) reflects tolerance for partially supported reasoning.

## 4 EXPERIMENTS

### 4.1 EVALUATION SETUP

#### 4.1.1 EVALUATED MODELS

We evaluate several instruction-tuned families of large language models (LLMs) across four size regimes on a unified diagnostic set of **468** cases. These models include: $\geq$**32B**: ChatGPT-4o-mini, DeepSeek-V3-250324 (DeepSeek-AI et al., 2024), Gemini (Team et al., 2023), **32B**: GLM-4-32B-0414 (GLM et al., 2024), Qwen3-32B (Yang et al., 2025), DeepSeek-R1-Distill-Qwen-32B (Guo et al., 2025), Qwen2.5-32B-Instruct (Hui et al., 2024), **14B**: Baichuan-M1-14B-Instruct, Qwen3-14B, DeepSeek-R1-Distill-Qwen-14B, Qwen2.5-14B-Instruct (Wang et al., 2025), $\sim$**8B**: Llama-3-8B-Instruct, Qwen3-8B, DeepSeek-R1-Distill-Qwen-7B, Qwen2.5-7B-Instruct (Grattafiori et al., 2024). All models adopt the same prompt templates and decoding hyperparameters to ensure a fair comparison across different architectures and parameter scales. Each model's output follows a strict format, which includes the label `[Final Diagnosis]` followed by three supporting evidence items. This ensures consistency in evaluation and allows for direct comparison of diagnostic accuracy and evidence quality.

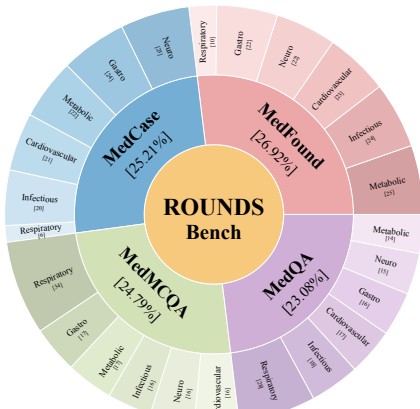

Figure 3: **Distribution of clinical cases in ROUNDS-Bench.** The benchmark integrates four source datasets: MedQA, MedMCQA, MedFound, and MedCase covering six major disease systems. Each segment shows the proportion of source contribution and category composition, ensuring balanced topic coverage across the benchmark.

### 4.1.2 EVALUATION PROTOCOL

**Task 1 (Full-Context).** In this task, the model is presented with the entire structured clinical case: $[\mathcal{P}, \mathcal{C}, H, PMH, PE, AE]$, and must return a diagnosis along with three supporting evidence items, $\hat{E} = \{e_1, e_2, e_3\}$. This task evaluates the model's performance when all relevant information is immediately available, establishing an upper bound for diagnostic accuracy.

**Task 2 (Active Evidence-Seeking).** In this task, the model is provided with only the initial context of patient demographics and chief complaint: $[\mathcal{P}, \mathcal{C}]$. The model interacts with the Standardized Patient Simulator (SP-sim) by issuing one discrete request per turn, chosen from the predefined action set $\mathcal{A}$, which includes options such as history-taking, physical examination, and auxiliary tests: $\mathcal{A} = \{\texttt{HPI}, \texttt{PMH}, \texttt{PE}, \texttt{LAB}(x), \texttt{IMG}(x), \texttt{FUNC}(x), \texttt{PANEL}(x), \texttt{STOP\_DIAG}\}$. The model must also decide when to issue the $\texttt{STOP\_DIAG}$ action, signifying that it is ready to make a final diagnosis. At this point, the model must ground all three supporting evidence items in the content retrieved during the multi-turn interaction. This setup mimics a more realistic clinical workflow, where the model incrementally gathers information to support its diagnostic reasoning.

### 4.1.3 STANDARDIZED PATIENT SIMULATOR (SP-SIM)

The SP-sim operates as a *strict request–response* engine, providing no unsolicited information. The model issues exactly one query per turn, with each response limited to the requested section (e.g., HPI, PE, lab tests, imaging). Responses are fully reproducible, ensuring that the evidence acquisition strategy is decoupled from diagnostic reasoning. This setup mirrors Objective Structured Clinical Examinations (OSCEs), where clinical encounters are simulated via interaction with standardized patients (SPs). The SP-sim enables assessment of both diagnostic accuracy and evidence-seeking strategies, making the benchmark sensitive to the strategic decisions in the "ask-examine-test-stop" cycle of clinical reasoning (see Fig. 2). By enforcing strict adherence to the request–response model, the SP-sim ensures that diagnostic conclusions are grounded in relevant, collected evidence, closely reflecting real-world clinical workflows. See Appendix B for detailed.

### 4.2 MAIN RESULTS

**Performance Degradation from Task 1 to Task 2** As shown in **Table 1**, all models across various parameter sizes demonstrate a consistent and significant performance drop when transitioning from **Task 1 (Full-Context Diagnosis)** to **Task 2 (Active Evidence-Seeking Diagnosis)**. Specifically, the

| Model | Task 1 | | | | Task 2 | | | | Drop | |
|---|---|---|---|---|---|---|---|---|---|---|
| | Exact | StrictEQ | GrdAcc$_{0.5}$ | EQ$_{0.5}$ | Exact | StrictEQ | GrdAcc$_{0.5}$ | EQ$_{0.5}$ | ΔE | ΔS |
| *≥32B Models* | | | | | | | | | | |
| Gemini-2.5-Pro | **65.2** | **78.8** | **73.7** | **87.1** | **49.36** | 42.52 | **55.77** | 57.05 | -15.84 | -36.28 |
| ChatGPT-4o-mini | 45.51 | 61.32 | 59.19 | 79.4 | 36.54 | 44.02 | 48.5 | 67.8 | -8.97 | -17.3 |
| Deepseek-v3-250324 | 59.19 | 75.43 | 70.0 | **87.1** | 46.15 | 50.21 | 54.3 | **70.0** | -13.04 | -25.22 |
| *˜32B Models* | | | | | | | | | | |
| GLM-4-32B-0414 | 47.22 | 62.82 | 59.62 | 77.03 | 35.68 | 39.53 | 44.98 | 60.04 | -11.54 | -23.29 |
| Qwen3-32B | **59.40** | **74.79** | **71.01** | **85.9** | **48.29** | 51.28 | **57.4** | **70.7** | -11.11 | -23.51 |
| DeepSeek-R1-Distill-Qwen-32B | 53.42 | 67.09 | 64.64 | 82.59 | 33.92 | 31.71 | 46.3 | 55.76 | -19.50 | -35.38 |
| Qwen2.5-32B-Instruct | 49.36 | 66.45 | 63.9 | 80.45 | 39.10 | 49.15 | 52.0 | 67.52 | -10.26 | -17.3 |
| *˜14B Models* | | | | | | | | | | |
| Baichuan-M1-14B | 53.42 | 66.45 | 64.21 | 80.45 | 44.87 | 44.87 | 54.17 | 67.52 | -8.55 | -21.58 |
| Qwen3-14B | **57.69** | **73.08** | **67.73** | **86.0** | **45.09** | **52.14** | **55.9** | **71.4** | -12.6 | -20.94 |
| DeepSeek-R1-Distill-Qwen-14B | 47.01 | 59.83 | 57.69 | 77.14 | 23.29 | 16.24 | 33.65 | 37.71 | -23.72 | -43.59 |
| Qwen2.5-14B | 42.52 | 58.76 | 56.62 | 77.2 | 33.97 | 39.53 | 45.8 | 63.0 | -8.55 | -19.23 |
| *˜8B Models* | | | | | | | | | | |
| Llama-3-8B | 36.75 | 48.08 | 47.56 | 66.5 | 24.79 | 22.86 | 34.4 | 45.2 | -11.96 | -25.22 |
| Qwen3-8B | **53.42** | **57.48** | **68.16** | **74.5** | **39.53** | **45.3** | **49.6** | **66.9** | -13.89 | -12.18 |
| DeepSeek-R1-Distill-Qwen-7B | 14.10 | 16.45 | 25.53 | 42.95 | 5.98 | 1.92 | 11.4 | 11.86 | -8.12 | -14.53 |
| Qwen2.5-7B | 36.97 | 48.93 | 49.86 | 70.9 | 23.29 | 19.23 | 34.8 | 46.7 | -13.68 | -29.7 |

Table 1: Performance of LLMs on **Task 1** (single-pass diagnosis) and **Task 2** (active evidence-seeking) across parameter scales. Values are percentages. Within each *size × task × metric* block, **best value(s)** are bolded (ties are co-bolded) and the runner-up is underlined. Δ columns report the absolute drop from Task 1 to Task 2 ($\Delta = $ Task 2 − Task 1; negative is worse).

**Exact match** score experiences an average decline of **12.5 percentage points** (median: 11.9), while the **StrictEQ** score drops even more dramatically by **26.1 points** (median: 23.4). This substantial degradation underscores the challenge of real-world clinical diagnostics, where models must actively seek out information under uncertain conditions and limited interaction turns, rather than working with complete case information, as in Task 1. The gap between the two tasks is particularly evident in models with smaller parameter sizes, such as those in the ˜8B category, where performance drops by as much as **–13.68 points** in Exact match and **–29.7 points** in StrictEQ. These findings reveal that active evidence-seeking introduces a level of complexity and uncertainty that significantly impacts model performance, particularly in scenarios where evidence must be gathered incrementally.

**Evidence Quality Degradation**    In addition to the drop in diagnostic accuracy, a more pronounced degradation occurs in the **Evidence Quality (EQ)** across all models in **Task 2**. As shown in the table, models generally exhibit a steeper decline in **StrictEQ** than in **Exact match**, suggesting that many models, while still able to make plausible diagnoses, struggle to ground their decisions in sufficient and discriminative evidence. For example, top models like **Qwen3-32B** and **Qwen3-14B** show a consistent drop in evidence quality despite maintaining strong diagnostic accuracy. **Qwen3-32B**, with the highest Exact match (48.3%), drops by **–36.28 points** in StrictEQ, indicating that it may generate accurate diagnoses but with limited evidence to support them. Meanwhile, **Qwen3-14B** achieves the best StrictEQ (52.1%) despite a lower Exact match, suggesting that it is more successful at gathering relevant evidence even if the final diagnosis is less precise. This trend highlights the importance of not only obtaining correct diagnoses but also ensuring that these diagnoses are supported by high-quality, relevant evidence—something that becomes more challenging when the model is required to actively seek and piece together information incrementally.

## 4.3 Exploring the Role of Model Scaling and Families in Task Performance

**Performance of Medical Models on Task 1 and Task 2.**    Figure 4 shows the performance of medical models on **Task 1 (Full-Context Diagnosis)** and **Task 2 (Active Evidence-Seeking Diagnosis)**. In **Task 1**, when the model receives complete clinical information, accuracy is generally high, especially for larger models like **Qwen2.5-32B-Instruct** and **Baichuan-M2-32B-Instruct**, with accuracies reaching 56%. However, in **Task 2**, where models must gather evidence step-by-

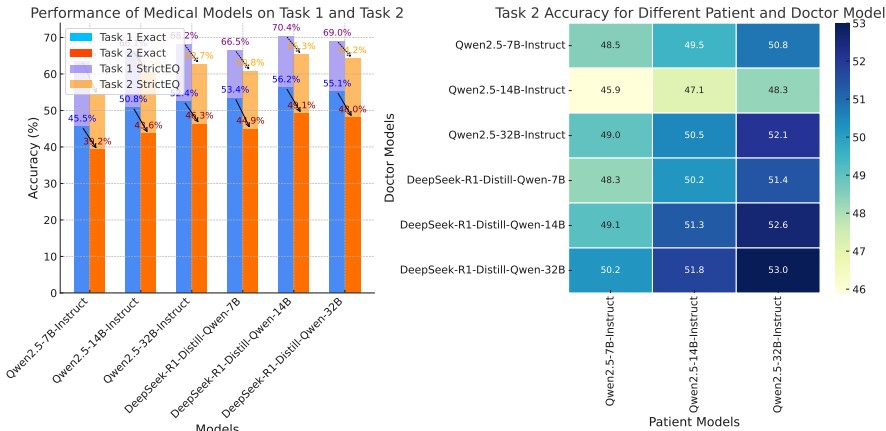

Figure 4: (Left) Performance of Medical Models on Task 1 and Task 2, comparing Exact and StrictEQ scores across models. (Right) Task 2 Accuracy for Different Patient and Doctor Models, showing slight accuracy improvement with larger models. Patient model size mainly controls data flow, with minimal impact on final accuracy.

step, accuracy drops by 5%-10% compared to **Task 1**. This reflects the challenge of **Task 2**, where the model's reasoning and evidence acquisition strategies are tested. Despite the drop, the increase in model size improves **StrictEQ** scores, showing that larger models can handle and integrate more information, though the improvement is modest.

**Task 2 Accuracy for Different Patient and Doctor Models.** Figure 4 illustrates the accuracy of **Task 2** for different **patient models** and **doctor models**. While larger **patient models**, such as **Qwen2.5-7B-Instruct**, **Qwen2.5-14B-Instruct**, and **Qwen2.5-32B-Instruct**, show slight improvements in accuracy (48.5%, 49.5%, 50.8%, respectively), their impact on final diagnostic accuracy is limited. The role of the patient model is primarily in **controlling data flow** rather than directly influencing diagnostic accuracy. In contrast, **doctor models**, such as the **DeepSeek-R1-Distill-Qwen** series, show a clearer increase in accuracy with larger models (from 50.2% to 53.0%). Thus, while patient model size controls data flow, the doctor model's size and reasoning ability are the main factors driving performance in **Task 2**.

**Impact of Model Family, Size, and Type on Performance.** Model scale and reasoning capability significantly affect performance. The **Qwen2.5** series, as non-think models, rely mainly on pre-trained patterns, leading to poorer performance in **Task 2**. Their accuracy drops substantially. In contrast, the **Qwen3** and **DeepSeek-R1-Distill-Qwen** series, as think models, perform better in multi-turn interactions, showing better stability and smaller drops in accuracy in **Task 2**, as they actively reason and gather evidence. As model size increases, particularly in the **32B** models, accuracy improves, with larger models demonstrating stronger reasoning and evidence integration in **Task 2**. However, the improvements are limited. Finally, think models, capable of active reasoning, are more adaptable in **Task 2**, while non-think models perform poorly in complex reasoning tasks. Overall, think models, especially larger ones, are crucial for better performance in **Task 2**. See Appendix E for detailed.

## 5 CONCLUSION

We present ROUNDS-Bench, a clinically-grounded benchmark designed to evaluate LLMs on the multi-turn diagnostic process by decoupling evidence acquisition from reasoning. Our findings reveal a critical performance gap: while state-of-the-art models are proficient with full context, they exhibit a stark degradation when required to actively seek evidence, a deficit most pronounced in the quality of their evidentiary support. As a reproducible platform with released code, ROUNDS-Bench provides a crucial tool for benchmarking and cultivating the active diagnostic competence essential for real-world clinical application.

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

ETHICS STATEMENT

This work does not present any significant ethical concerns. All datasets used in our research, such as [e.g., ImageNet, CIFAR-100], are publicly available and well-established academic benchmarks. The data are anonymized and does not contain personally identifiable information. Our proposed method, which focuses on [e.g., improving model efficiency and accuracy], does not have foreseeable direct negative societal impacts. No human subjects were involved in our experiments.

STATEMENT ON LLM USAGE

During the preparation of this work, we used LLMs, such as ChatGPT, for assistance with language editing, grammar correction, and improving readability. All core ideas, methodologies, experimental designs, results, and conclusions were originally conceived and articulated by the human authors. We have carefully reviewed and edited all text generated or modified by LLMs and take full responsibility for the final content of this paper. LLMs were not listed as authors.

## A  DATASET CONSTRUCTION AND PRE-PROCESSING

**Sources and balancing.**  ROUNDS-Bench integrates four widely used resources—MedQA, MedMCQA, MedFound, and MedCase—and standardizes all instances into six major clinical systems to ensure balanced coverage. Table 2 summarizes the final distribution: each system contributes **78** cases, totaling **468** structured instances across the four sources. This design enables like-for-like comparison across organ systems and model families.

**Stage I: Diagnostic-only filtering.**  To isolate diagnostic reasoning from other question types (e.g., mechanism, treatment, "next best step"), we apply a two-step screen (Fig. 5): (*i*) a *diagnosis-type filter* that checks whether the item requires predicting a disease/diagnosis from a clinical vignette; and (*ii*) a *diagnosis-term filter* that verifies the candidate answers are *disease entities* rather than symptoms, tests, or therapies. An item is retained *only if* it passes both prompts (Yes → Yes), ensuring that downstream evaluation focuses on diagnostic endpoints.

**Stage II: Schema-constrained structuring.**  For retained items, we convert free-text vignettes into a six-section clinical record—*Patient Information*, *Chief Complaint*, *History of Present Illness*, *Past Medical History*, *Physical Examination*, and *Auxiliary Examination*—using a schema-constrained prompt (Fig. 6). The prompt enforces (a) fixed section headers and slot discipline, (b) no diagnosis leakage, and (c) explicit "None" for missing fields. This step makes evidence explicit and decouples *evidence acquisition* from subsequent *diagnostic reasoning*.

**Stage III: Structural validation and fabrication check.**  A lightweight validator (Fig. 7) audits the structured record against the original text. It requires (1) the presence of all six sections, (2) no fabricated details ("None" allowed when absent), and (3) faithful formatting; the outcome is a strict binary (yes/no). Only records passing the validator enter the benchmark to guarantee format consistency and evidence faithfulness.

**Illustrative structured case (no diagnosis).**  Figure 8 presents an example record after structuring: demographics, a concise chief complaint with timeline, focused HPI, vitals and exam, and multimodal auxiliary studies (imaging, laboratory, ECG, hemodynamics, biopsy). No diagnostic label is included at this stage, preserving the intended separation between *evidence* and *diagnosis* for both Task 1 (full-context) and Task 2 (active evidence-seeking).

**System-level categorization.**  Finally, diseases are mapped to six clinical systems using a transparent rule-based prompt (Fig. 9). The instruction returns a JSON tuple {`primary_diagnosis`, `category`}, aligning each case to one of the six systems or `Other` when appropriate. This categorization underpins the balanced distribution reported in Table 2 and supports per-system analyses in the experiments.

**Prompt 1 — Diagnosis-Type Filter**

You are a medical board exam assistant. Please determine whether the following question is a diagnosis-type question.
A diagnosis-type question requires the examinee to determine the most likely diagnosis or disease based on the clinical presentation. It does not ask about etiology, treatment, mechanisms, or next steps.
Please answer with only one word: "Yes" or "No".
**Question stem:** `<question_text>`
**Options:** `<options_text>`
**Answer:**

**Prompt 2 — Diagnosis-Term Filter**

You are a medical expert. Please determine whether the following terms are medical diagnoses (names of diseases or conditions), rather than symptoms, tests, mechanisms, or treatments. Please answer with only one word: "Yes" or "No".
**Options:** `<options_text>`

Figure 5: Side-by-side prompts used for diagnostic-only filtering (Stage 1).

## B  STANDARDISED PATIENT SIMULATOR (SP-SIM)

**Simulator role and model.**    We instantiate the patient side of the dialogue as a *standardised patient* driven by **Qwen2.5-7B-Instruct**. The simulator is *strictly reactive*: it answers only to the clinician's most recent utterance and *never* discloses unsolicited information or the final diagnosis. Default decoding settings are listed in Table **??** to ensure determinism and reproducibility.

Table 2: Distribution of clinical system categories across ROUNDS-Bench datasets. Full category names: CV (Cardiovascular), RS (Respiratory), GH (Gastro-Hepatobiliary), NS (Neurological), ID (Infectious Diseases), MRGU (Metabolic, Renal & Genitourinary).

| Category | MedQA | MedMCQA | MedFound | MedCase | Total |
|----------|-------|---------|----------|---------|-------|
| CV | 17 | 15 | 25 | 21 | 78 |
| RS | 28 | 34 | 10 | 6 | 78 |
| GH | 16 | 18 | 20 | 24 | 78 |
| NS | 15 | 16 | 22 | 25 | 78 |
| ID | 18 | 16 | 24 | 20 | 78 |
| MRGU | 14 | 17 | 25 | 22 | 78 |
| **Total** | **108** | **116** | **126** | **118** | **468** |

**Opening utterances.**    To initiate the encounter, the clinician uses one of **15** templated openings (Fig. 10); these variants reduce prompt overfitting to any particular phrasing. Upon the first opening, the SP-sim must return **exactly** the `1.Patient Information` and `2.Chief Complaint` sections from the structured record produced in the previous stage, without any inference or additional details.

**Gated information release.**    After the opening, the SP-sim releases content only when the clinician explicitly requests one of the allowed modules—`History of Present Illness`, `Past Medical History`, `Physical Examination`—or requests a specific test block (`Laboratory Tests: ...`, `Imaging Studies: ...`, `Functional Tests: ...`, `Specialized Panels: ...`). For each request, the simulator *echoes verbatim* the corresponding section or test result from the case record. If a requested test is not present, it responds with `"This test was not performed yet."` This gating policy (Fig. 11) enforces the decoupling between *evidence acquisition* and *diagnostic reasoning* and prevents leakage of the final diagnosis.

---

**Prompt: Schema-Constrained Structuring**

You are a clinical documentation expert. Convert any form of clinical vignette or patient record into a standardized medical record format. Do NOT infer or include any diagnosis.
Use this exact structure:
**1. Patient Information**
- [Sex, Age] (or "None")

**2. Chief Complaint**
- [Primary symptom] + [Duration] (or "None")

**3. History of Present Illness**
- Progression: [Chronological illness course]
- Accompanying symptoms: [Comma-separated symptoms]
(Use "None" if not available)

**4. Past Medical History**
- [Relevant history] (or "None")

**5. Physical Examination**
- Vital signs: [Temperature, HR, RR, BP...] (or "None")
- [System-specific findings] (or "None")

**6. Auxiliary Examination**
- (1) Imaging test: [Findings] (or "None")
- (2) Laboratory tests: [Key abnormal results] (or "None")
- (3).....

Repeat back only the structured output.
Please convert the following clinical vignette or case into a structured medical record.
Do NOT include any diagnosis results. Fill "None" for missing fields. The text may be a clinical note or exam question.

**Case:** question_text

Please strictly follow this format, but leave out the final diagnosis:
```
1.Patient Information
2.Chief Complaint
3.History of Present Illness
4.Past Medical History
5.Physical Examination
6.Auxiliary Examination
```

---

Figure 6: Figure 6: Schema-constrained prompt used to convert free-text vignettes into a six-section clinical record (no diagnosis). The schema enforces fixed headers, no leakage, and explicit "None" for missing fields.

| Setting | Default Value |
| --- | --- |
| Model | Qwen2.5-7B-Instruct |
| Temperature | 0.7 |
| Top-p | 0.8 |
| Top-k | 20 |
| Repetition penalty | 1.0 |
| Max new tokens | 1024 |

Table 3: Default decoding and control settings for the standardised patient.

## C  DOCTOR AGENT (CLINICIAN)

**Role and interaction protocol.** The doctor agent executes clinical decision-making under two settings: **Task 1** (full-context) and **Task 2** (active evidence-seeking). In Task 1, the agent consumes

---

**Prompt: Structure & Fabrication Check**

You are a critical evaluator of medical documentation quality.
Your task is to determine whether the following structured medical record is a faithful and properly formatted transformation of the original clinical case description.
**Evaluation Criteria:**

1. The record must contain all 6 required sections with their respective content:

   - Patient Information
   - Chief Complaint
   - History of Present Illness
   - Past Medical History
   - Physical Examination
   - Auxiliary Examination

   *Note: Minor variations in section titles (e.g., spacing, punctuation, casing) are acceptable as long as the structure is clearly preserved.*

2. The structured record must **not fabricate any content**. All included details must be:

   - Explicitly stated in the original case, **or**
   - Clearly implied with no assumptions beyond clinical description.

3. If any section lacks source information, using `"None"` is acceptable.

**Original Case:** `original_text`
**Structured Medical Record:** `formatted_record`
Please assess strictly but reasonably.
**Answer only with** `yes` (fully valid) **or** `no` (any fabrication, omission, or structural failure).

Figure 7: Validation prompt for auditing the structured record against the original text: all six sections present, no fabrication ("None" allowed), and faithful formatting. Output is a strict yes/no.

| Model (Doctor Agent) | Size | Temp | Top-p | Top-k |
|---|---|---|---|---|
| Gemini | ≥32B | 1.0 | 0.95 | 40 |
| ChatGPT-4o-mini | ≥32B | 1.0 | 0.95 | 40 |
| DeepSeek-V3-250324 | ≥32B | 1.0 | 0.95 | 40 |
| GLM-4-32B-0414 | 32B | 1.0 | 0.95 | 40 |
| Qwen3-32B | 32B | 0.6 | 0.95 | 20 |
| DeepSeek-R1-Distill-Qwen-32B | 32B | 0.6 | 0.95 | 20 |
| Qwen2.5-32B-Instruct | 32B | 0.7 | 0.80 | 20 |
| Baichuan-M1-14B-Instruct | 14B | 1.0 | 0.95 | 40 |
| Qwen3-14B | 14B | 0.6 | 0.95 | 20 |
| DeepSeek-R1-Distill-Qwen-14B | 14B | 0.6 | 0.95 | 20 |
| Qwen2.5-14B-Instruct | 14B | 0.7 | 0.80 | 20 |
| Llama-3-8B-Instruct | 8B | 1.0 | 0.95 | 40 |
| Qwen3-8B | 8B | 0.6 | 0.95 | 20 |
| DeepSeek-R1-Distill-Qwen-7B | 8B | 0.7 | 0.80 | 20 |
| Qwen2.5-7B-Instruct | 8B | 0.7 | 0.80 | 20 |

Table 4: Doctor-side models and their official default decoding settings (greedy or sampling as specified).

the entire structured record and must output a single `[Final Diagnosis]` with exactly three supporting items. In Task 2, the agent interacts turn-by-turn with the SP-sim (§**??**), *issuing one action per turn* from the allowed action set (history, past history, physical exam, or a specific test request) and may stop with `[Final Diagnosis]`. All supporting evidence must be *verbatim retrievable* from content obtained so far; repeated or unavailable tests are disallowed, and the dialogue is capped at **10 turns**.

**Evaluated doctor models and decoding.** We run **15** instruction-tuned LLMs as the doctor agent (Table 4). To ensure strict comparability, all models share deterministic decoding and identical

---

**Structured Case Example**

**1. Patient Information**
- Male, 44

**2. Chief Complaint**
- Chills for 3 days and arthralgias in the knees and hips (preceded by several days of unproductive cough and headache)

**3. History of Present Illness**
- Progression: Unproductive cough and headache preceded chills and arthralgias. One week before presentation, he was treated with a macrolide antibiotic and an NSAID.
- Accompanying symptoms: Cough, Headache, Chills, Arthralgias

**4. Past Medical History**
- Smoking history (None otherwise)

**5. Physical Examination**
- Vital signs: Temperature 38.5 °C, Heart Rate 113/min, Blood Pressure 126/64 mmHg, Oxygen Saturation 98% on room air
- Findings: No pericardial rub or crackles; epigastric tenderness

**6. Auxiliary Examination**
- (1) Imaging test: Chest radiograph showed mild peribronchial cuffing. Transthoracic echocardiography revealed preserved LV function, a 9-mm pericardial effusion, and slight IVC dilation. Coronary CT excluded obstructive disease. Cardiac MRI demonstrated myocardial edema with multifocal subepicardial and subendocardial late gadolinium enhancement and pericardial inflammation.
- (2) Laboratory tests: WBC $13.4 \times 10^3/\mu$L, CRP 16.9 mg/dL, ESR 95 mm/h, high-sensitivity troponin T 656.2 ng/L; differential count showed no eosinophilia. Blood cultures, serology, and PCR for pathogens negative; vasculitis-associated autoantibodies absent.
- (3) Electrocardiogram: Sinus tachycardia with first-degree AV block (PQ 210 ms).
- (4) Right heart catheterization: Cardiac Index 1.65 L/min/m$^2$, mean PCWP 34 mmHg, LVEDP 29 mmHg.
- (5) Endomyocardial biopsy: Eight specimens obtained from the left ventricle.

---

Figure 8: Illustrative example of a structured case produced by the schema in Fig. 6. The record contains demographics, chief complaint with timeline, focused HPI, exam with vitals, and auxiliary studies. No diagnosis is included.

prompt scaffolds (Figs. 12–13). Unless otherwise specified, temperature is zero (greedy), and we fix the random seed.

**Output discipline.** For both tasks, the agent must begin the final answer with the exact tag `[Final Diagnosis]` and list three items as confirmations (no free-form diagnosis without evidential grounding). In Task 2, for non-terminal turns the agent outputs *(chosen action): (brief rationale)* to make the acquisition policy auditable without revealing the final diagnosis.

## D    EVALUATION

| Setting | Default Value |
|---|---|
| Model | Qwen2.5-Coder-32B-Instruct |
| max_new_tokens | 10 |
| temperature | 0.7 |
| top_p | 0.95 |
| top_k | 20 |
| do_sample | false |

Table 5: Default settings for the evaluator

---

**Prompt: System-Level Categorization** fonttitle

You are a medical assistant with extensive clinical knowledge. Your task is to classify each disease name into one of the following six major clinical system categories. If a disease does not clearly belong to any of these categories, label it as **"Other"**.

**Use the following classification rules:**

**1. Cardiovascular System**
- Includes: Acute coronary syndrome, heart failure, arrhythmias (e.g., atrial fibrillation), hypertensive emergencies, aortic dissection, pericarditis, etc.

**2. Respiratory System**
- Includes: Asthma, COPD, pneumonia, pulmonary embolism, spontaneous pneumothorax, hemoptysis-related diseases, etc.

**3. Gastro-Hepatobiliary System**
- Includes: Upper or lower GI bleeding, appendicitis, cholecystitis, pancreatitis, liver cirrhosis and complications, inflammatory bowel disease, abdominal pain, diarrhea, etc.

**4. Neurological System**
- Includes: Ischemic stroke, TIA, seizures/epilepsy, subarachnoid hemorrhage, headaches, dizziness/vertigo, migraine, CNS infections, etc.

**5. Infectious Diseases**
- Includes: Bacterial meningitis, urinary tract infections, community or hospital-acquired pneumonia, skin and soft tissue infections, early sepsis, tropical diseases (e.g., dengue, malaria), etc.

**6. Metabolic, Renal & Genitourinary System**
- Includes: Diabetes mellitus (DKA, HHS), hypoglycemia, thyroid disorders (hyper/hypothyroidism), electrolyte disorders (e.g., hyponatremia, hyperkalemia), acute kidney injury, kidney stones, urinary tract diseases, etc.

If a disease does not fit into any of the above six categories, classify it as: **Other**

Please return your answer in JSON format, like this:
```
{"primary_diagnosis": "Pneumonia", "category": "Respiratory System"}
```

Figure 9: Rule-based categorization prompt that maps diseases to one of six clinical systems (or `Other`) and returns a JSON tuple {`primary_diagnosis`, `category`}. This supports the balanced per-system distribution in Table 2.

**Evaluator model and protocol.** We use **Qwen2.5-32B-Instruct** as an external evaluator to score model outputs under both tasks. For each case, the evaluator receives the gold diagnosis, the model's final diagnosis (with the required tag), and the structured case content needed to verify evidence items. Deterministic decoding is enforced to guarantee reproducibility across runs.

**Endpoints and scoring.** We report two endpoints: *Final Diagnosis Accuracy* and *Evidence Quality*. Accuracy is scored on a 3-level scale {0,1,2}: **0** incorrect (different disease family), **1** partially correct (right family, wrong subtype/term), **2** fully correct (including synonyms/subtypes). Evidence Quality is also scored on {0,1,2}: **2** all three evidence items are supported and consistent; **1** one–two items supported or partially reasonable; **0** unsupported/contradictory. Figures 14 and 15 show the prompts used to elicit *single-integer* decisions.

**Determinism and guardrails.** Default evaluator settings are listed in Table 5: temperature 0 with fixed seed, greedy decoding, and a strict output contract (single integer, no extra text). The evaluator only judges (i) the [Final Diagnosis] string against the gold label for Accuracy and (ii) whether the three evidence bullets are *verbatim supported* by the visible case content for Evidence Quality. No external knowledge is used, and diagnosis leakage is irrelevant at evaluation time.

**Opening Utterances (15 Templates)**

- "Hello, what brings you in today?"
- "Can you tell me the reason for your visit?"
- "Let's begin — have you noticed anything unusual lately?"
- "What symptoms have you been experiencing?"
- "What seems to be the problem today?"
- "What kind of health concerns are you having recently?"
- "Please describe any discomfort or issues you've been having."
- "Is there anything in particular that's been bothering you?"
- "Tell me what's going on — when did it start?"
- "Are you here because of any new or ongoing symptoms?"
- "What made you decide to come see a doctor today?"
- "We'll start with your concerns — what would you like to discuss?"
- "Have you experienced any recent changes in your health?"
- "Can you walk me through your current symptoms?"
- "What's the main reason for your appointment today?"

Figure 10: Fifteen interchangeable opening utterances used by the clinician to start the encounter. On the first turn, the SP-sim responds with `1.Patient Information` and `2.Chief Complaint` verbatim from the structured case.

## E   VISUALIZATION OF MODEL PERFORMANCE

This section presents the visualizations that compare the performance of various models across different tasks, model sizes, and model types. 16The following plots are provided for detailed analysis:

1. **Exact Accuracy Comparison for Model Families (Qwen vs DeepSeek):** Line plot showing exact accuracy for both Qwen and DeepSeek models. This plot highlights how the model families perform on **Task 1** and **Task 2** in terms of exact accuracy.

2. **Drop in Exact Accuracy for Model Families (Qwen vs DeepSeek):** Bar plot illustrating the drop in accuracy from **Task 1** to **Task 2**. The plot shows how accuracy decreases when models transition from single-pass diagnosis to active evidence-seeking.

3. **Heatmap of Task 1 and Task 2 Performance for Different Models:** A heatmap visualizing the performance of all models across **Task 1** and **Task 2**. The heatmap provides a quick overview of the exact and StrictEQ accuracy for each model.

4. **Exact Accuracy for Models with Different Sizes:** Box plot comparing the **Exact Accuracy** for **Task 1** and **Task 2** across different model sizes. This plot shows how model size impacts performance in terms of exact accuracy.

5. **Drop in Exact Accuracy Across Different Model Sizes:** Violin plot showing the distribution of the drop in exact accuracy across different model sizes. This helps in understanding the variance in performance drop for different model sizes.

6. **Comparison of Task 1 and Task 2 StrictEQ for Models:** Stacked bar plot comparing **StrictEQ accuracy** for **Task 1** and **Task 2**. The plot demonstrates how well models can justify their diagnosis using evidence.

7. **Exact Accuracy for Think vs Non-think Models (Qwen vs DeepSeek):** Scatter plot comparing **think** and **non-think** models in **Task 1** and **Task 2**. This plot illustrates the difference in performance between models with active reasoning (think models) and those relying on predefined responses (non-think models).

---

**S2. SP-sim Policy Prompt (Patient Role)**

You are simulating a real patient during a clinical consultation. You will **ONLY** answer based on this record, and **ONLY** respond to the doctor's most recent statement.

—— **Case Record** ——
`formatted_record`

**Opening Behavior:**
When the consultation begins, the doctor will ask a general opening question (e.g., "What brings you in today?" or "Where do you feel unwell?").
In your first response, you must present the following two sections **exactly as written in the case record**:
- **1.Patient Information**
- **2.Chief Complaint**

**After the opening, respond ONLY when the doctor requests a specific module or test.**

**Allowed Module Responses:**
- [History of Present Illness]
- [Past Medical History]
- [Physical Examination]

**Allowed Examination Requests (you will recognize them like this):**
- [Laboratory Tests: ...]
- [Imaging Studies: ...]
- [Functional Tests: ...]
- [Specialized Panels: ...]

**How to respond:**

- Only respond if the requested test **is present in the case record**.
- Return the exact predefined result from the record.
- If a test is **not included in the case**, reply with:
  `"This test was not performed yet."`
- **Do NOT** invent any additional information.
- **Do NOT** include or hint at the final diagnosis at any time.

**Response Format:**
`[Module Name]:  Full original text from that section`
`[test_name]:  Full original test_result from that section`

---

Figure 11: Policy prompt for the standardised patient driven by Qwen2.5-7B-Instruct. The simulator is reactive, releases only requested modules/tests, echoes verbatim from the record, and never reveals the diagnosis.

8. **Drop in Accuracy for Think vs Non-think Models:** Area plot showing the drop in accuracy between **Task 1** and **Task 2** for **think** and **non-think** models. This plot demonstrates the impact of model type on performance degradation across tasks.

9. **Heatmap of Performance for Think vs Non-think Models:** Heatmap comparing the performance of **think** models on **Task 1** and **Task 2**. This visualization provides insight into how models with reasoning capabilities handle complex tasks.

These plots offer a detailed comparison across model families, model sizes, and model types, helping to identify the strengths and weaknesses of each model under different task conditions.

## F  COMPLETE RESULTS

**Overview.** We provide exhaustive breakdowns complementary to the main tables: (i) a **per-dataset** view across the four sources (MedQA, MedMCQA, MedFound, MedCase), and (ii) a **per-**

---

**D1. Task 1 — Full-Context Clinician Prompt**

You are an experienced senior clinician at a top-tier tertiary hospital. Your task is to carefully analyze structured patient records and provide the most accurate final diagnosis based on the clinical information.

You must follow strict clinical reasoning and adhere to the output format and diagnostic criteria described below.

**Output Format**
- When reaching a final diagnosis, **YOU MUST** start the response with:
```
[Final Diagnosis] [Diagnosis Name].  Confirmed by:
1.  ...
2.  ...
3.  ...
```

- You **MUST** include the exact tag `[Final Diagnosis]` with brackets — do not rephrase, omit, or replace it.

---

Figure 12: Clinician prompt for **Task 1** (full context). The agent reads the complete structured record and must output `[Final Diagnosis]` followed by exactly three evidential items, using the mandatory tag verbatim.

**system** view across the six clinical categories. Each grid contains **15 subplots**, one per doctor model. Bars are color-coded as T1-Exact, T1-StrictEQ, T2-Exact, T2-StrictEQ. All scores are shown on a normalized $0\sim1$ scale for direct visual comparison.

**Per-dataset results.** Figure 17 reports, for every model, Task 1 vs. Task 2 performance on each source. Across subplots, *Task 2* bars are consistently lower than *Task 1*, and the gap is typically wider for **StrictEQ** (red vs. orange) than for **Exact** (green vs. blue).

**Per-system results.** Figure 18 shows Task 1 vs. Task 2 across the six disease systems for each model. Again, *Task 2* losses are more pronounced on **StrictEQ**, indicating that models often reach plausible diagnoses while providing weaker or fewer case-supported findings under interactive constraints.

1080
1081
1082
1083
1084
1085
1086

**D2. Task 2 — Active Evidence-Seeking Clinician Prompt**

You are an experienced senior clinician at a top-tier tertiary hospital. Your goal is to gather only the necessary clinical information to reach a diagnosis.

You may choose **ONE** action per turn from:

- `[History of Present Illness]`
- `[Past Medical History]`
- `[Physical Examination]`
- Request `[Laboratory Tests:  test_name]`
- Request `[Imaging Studies:  test_name]`
- Request `[Functional Tests:  test_name]`
- Request `[Specialized Panels:  test_name]`
- `[Final Diagnosis]`

**Your output format should be:**
```
(one of the actions):  (your reasoning process for choosing the next action)
```

**Output Format for Final Diagnosis:**

- When reaching a final diagnosis, **YOU MUST** start the response with:
  ```
  [Final Diagnosis] [Diagnosis Name].  Confirmed by:
  1.  ...
  2.  ...
  3.  ...
  ```
- You **MUST** include the exact tag `[Final Diagnosis]` with brackets — do not rephrase, omit, or replace it.

**Note:**

- To improve diagnostic efficiency, please perform tests only when necessary for diagnosis.
- You may only request **one specific test per turn**.
- **Do NOT** repeat tests or other modules.
- When confident, issue a `[Final Diagnosis]`.
- You must complete the diagnosis within a maximum of **10 turns**.
- If a test or module returns `"This test was not performed yet."`, you **MUST NOT** request it again.
- If your previous request failed (e.g., returned `"This test was not performed yet."`), consider alternatives or proceed to diagnosis if sufficient.
- For each turn, check previous requests before proposing the next action.
- **Do not re-request** the same test, even if it previously failed.

Figure 13: Clinician prompt for **Task 2** (active evidence-seeking). Each turn, the agent selects exactly one action (module or a specific test), avoids repetitions and unavailable tests, and stops with `[Final Diagnosis]` within 10 turns. For non-terminal turns it outputs `(action): (brief rationale)`; final outputs must include the mandatory tag and three supporting items.

---

**E1. Accuracy Evaluation Prompt (0/1/2)**

You are a senior medical expert tasked with evaluating whether the AI model's diagnosis is accurate. Score according to the criteria below:

**[Scoring Criteria]**

- **0**: Completely incorrect (a different disease category)
- **1**: Partially correct (correct disease category, but wrong subtype)
- **2**: Completely correct (including synonyms or correct subtype)

**[Output Requirements]**
Return only a single integer score (0, 1, or 2). **Do NOT** include any other text.

---

Figure 14: Prompt used by the external evaluator to score **Final Diagnosis Accuracy** on a 0/1/2 scale. The evaluator returns a *single integer* only. This prompt is used identically for **Task 1** and **Task 2**.

---

**E2. Evidence Quality Evaluation Prompt (0/1/2)**

You are a medical evidence expert tasked with evaluating how well the diagnostic evidence provided by the model matches the clinical case information. Please assign a score based on the following criteria:

**[Scoring Criteria]**

- **2**: All 3 evidence points are clearly supported in the case and fully consistent with the diagnosis
- **1**: 1–2 evidence points are supported, or there is partial reasonable inference
- **0**: Evidence points contradict the case or are not supported at all

**[Output Requirement]**
Only return a single integer score (0, 1, or 2). **DO NOT** include any other text.
**[Important]**

- Only evaluate the model's evidence points (usually 1–3)
- Evidence must be specific, verifiable, and clearly stated

---

Figure 15: Prompt used by the external evaluator to score **Evidence Quality** on a 0/1/2 scale. The evaluator checks that the model's three evidence items are *verbatim supported* by the case and consistent with the diagnosis, and returns a *single integer* only. This prompt is used identically for **Task 1** and **Task 2**.

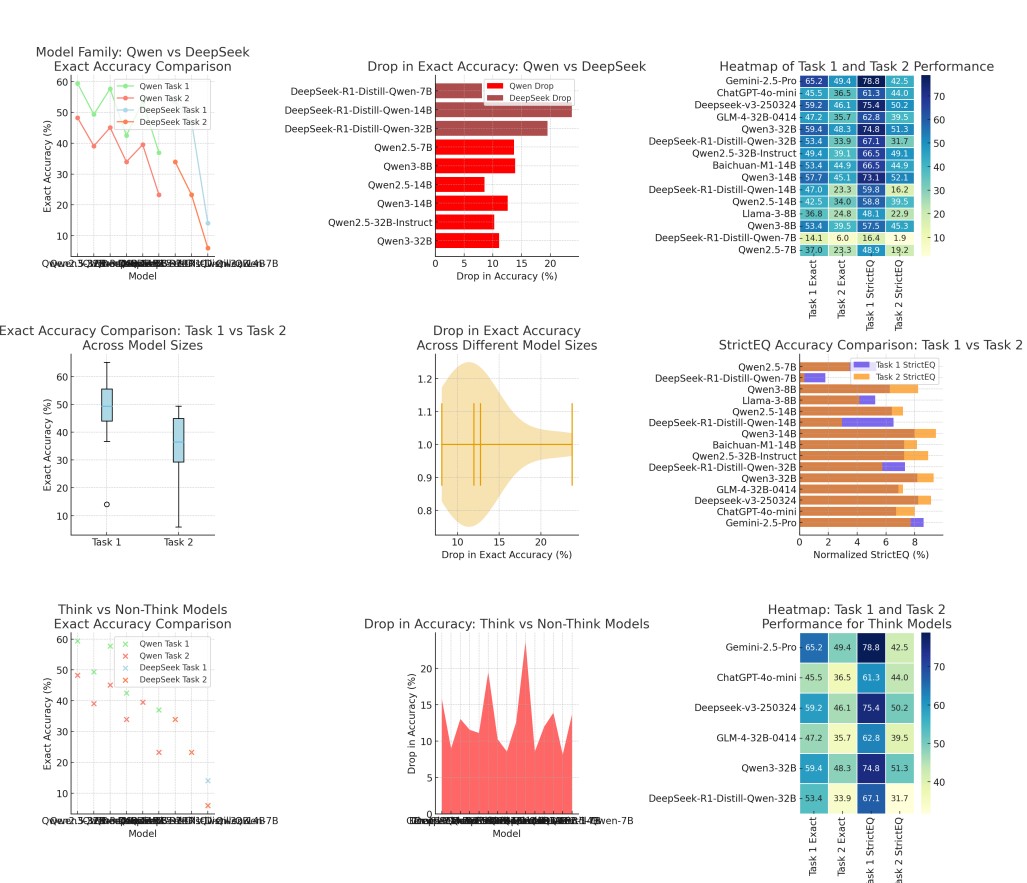

Figure 16: Visual comparisons of model performance across different tasks, model families, model sizes, and types. The plots include Exact Accuracy comparisons, drop in performance from Task 1 to Task 2, heatmaps, and analysis for think vs non-think models. These visualizations provide insights into the impact of model family, size, and reasoning capabilities on performance.

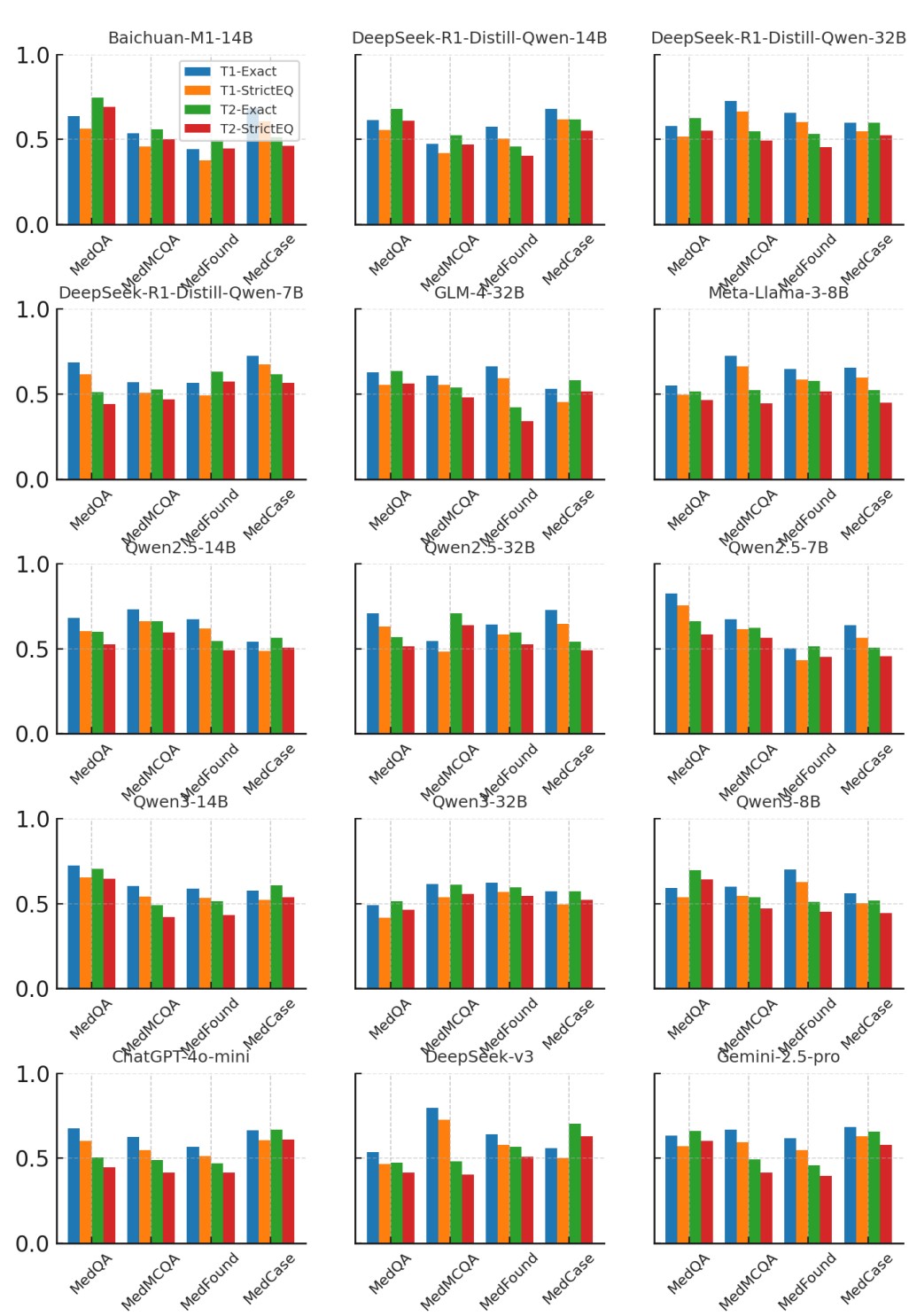

Figure 17: **Per-dataset breakdown across models (15 subplots).** Each subplot corresponds to one doctor model and plots results on *MedQA*, *MedMCQA*, *MedFound*, and *MedCase*. Bars are color-coded: T1-Exact, T1-StrictEQ, T2-Exact, T2-StrictEQ; the $y$-axis is normalized to $[0,1]$. The grid shows that Task 2 degrades most on evidence quality (red vs. orange), with source-dependent variability in the gap.

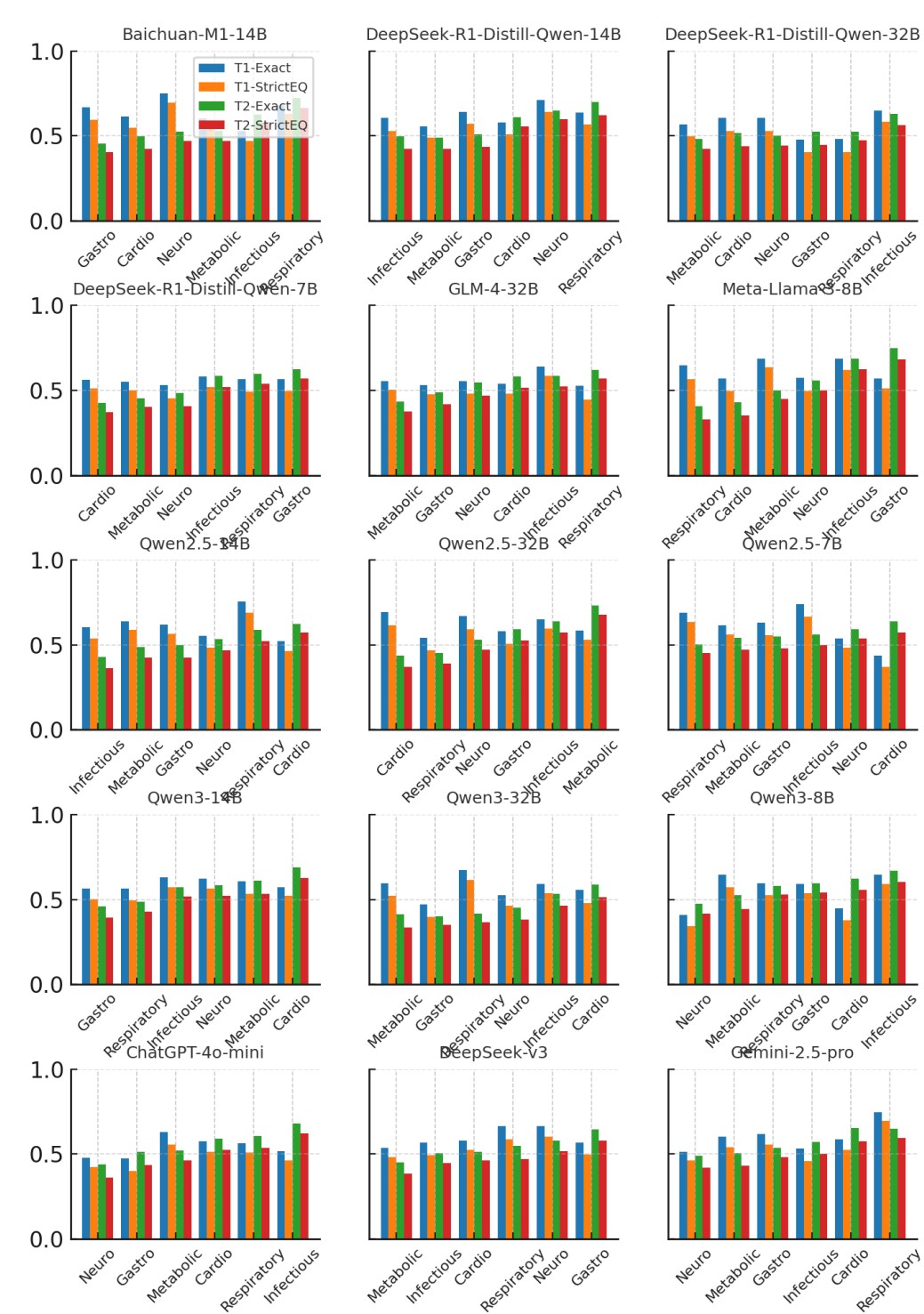

Figure 18: **Per-system breakdown across models (15 subplots).** Each subplot corresponds to one doctor model and reports performance across the six disease systems, with the same color coding (T1-Exact, T1-StrictEQ, T2-Exact, T2-StrictEQ) and $y$-axis normalization. Task 2 losses are again larger for StrictEQ, and the gap size varies by system, highlighting where interactive evidence acquisition is most demanding.

