# OpenReview forum: "Transitioning from Full-Context to Active Evidence-Seeking Evaluation: A Novel Benchmark for Real-World Artificial Intelligence Assisted Medical Diagnosis"
_ICLR.cc/2026/Conference — ICLR 2026 Conference Withdrawn Submission_

### Official Review · Reviewer_DMpG · 2025-11-01

**Soundness:** 3
**Presentation:** 2
**Contribution:** 3
**Rating:** 6
**Confidence:** 3

**Summary:**

The paper introduces ROUNDS‑Bench, an evaluation framework and dataset for active, multi‑turn medical diagnosis. Unlike static, full‑context benchmarks, Task 2 of ROUNDS‑Bench reveals only demographics and the chief complaint; a model must proactively request history/exam/tests, decide when to stop gathering information, and then deliver a final diagnosis with three supporting evidence items. Task 1 is a standard full‑context setting that serves as an upper bound. The benchmark integrates and restructures cases from MedQA, MedMCQA, MedFound, and MedCase into a 468‑case corpus balanced across six clinical systems (78 each). A standardized patient simulator (SP‑sim) enforces a strict request‑response protocol so that only requested information is returned. Evaluation focuses on two endpoints: (i) diagnosis accuracy (exact and graded) and (ii) Evidence Quality (EQ), judged by an external LLM with deterministic prompts. Across 15 models, the paper reports consistent degradation from Task 1 to Task 2, with larger drops in EQ than in exact accuracy, arguing that today’s models struggle more with evidence gathering/grounding than with passive reasoning.

Key elements shown in the paper:

* Framework & pipeline (dataset structuring -> Task 1 -> Task 2 -> evaluation).
* Action set for Task 2, including HPI, PMH, PE, LAB(x), IMG(x), FUNC(x), PANEL(x), and STOP DIAG, with one action per turn and max 10 turns.
* Dataset balance (78 cases/system; total 468).
* Main results with drops from Task 1 to 2.

**Strengths:**

* Well‑motivated shift from static to active diagnostic evaluation; clear OSCE inspiration and aligned design.
* Clean task formalization with explicit action space and stop decision, enabling strategy evaluation under interaction constraints.
* Balanced, structured dataset (468 cases; 6 systems × 78), with transparent structuring/validation steps.
* Two‑endpoint metric (accuracy and EQ) directly captures diagnostic correctness and evidentiary grounding; consistent performance drops in Task 2 substantiate the paper’s thesis.
* Reproducibility orientation: fixed prompts, strict output tags, and a deterministic SP‑sim policy.

**Weaknesses:**

1. Evaluator dependence & potential bias. Both endpoints are judged by one LLM (Qwen2.5‑32B‑Instruct) with no human study, no cross‑model adjudication, and no error analysis of the judge. This is especially problematic for EQ, which demands checking whether evidence is "verbatim supported"; borderline paraphrases could be mis‑scored. A small human‑validated subset and agreement statistics would strengthen claims.

2. Decoding / protocol inconsistencies. The paper claims shared decoding hyperparameters across models, yet Table 4 enumerates different temps/top‑p/top‑k; elsewhere it states temperature=0 unless noted. The actual settings used for Table 1 must be clarified; otherwise fairness and reproducibility are in question.

3. Simulator realism gaps. The SP‑sim does not model ordering new tests, cost, time, contraindications, or adverse effects; it only releases pre‑encoded case sections/tests and replies "not performed yet" otherwise. This bounds the action space and may over‑constrain the problem relative to real OSCEs. Incorporating costed test catalogs, availability, and contraindication rules would enhance realism.

4. Limited process analytics. The paper qualitatively mentions failure modes (redundant requests, premature closure), but provides no quantitative breakdown (e.g., per‑case turn counts, fraction of invalid/duplicate test requests, or how often models stop too early/late). Such analyses would help target method development.

5. Editing artifacts. "Table ??," an ethics statement that references ImageNet/CIFAR (not used), and patient‑model size inconsistency (Appendix B fixes the simulator to Qwen‑7B; Table 4 varies the patient model) detract from presentation quality.

6. External validity & baselines. No human baseline (e.g., physician residents) or non‑LLM agent baselines (e.g., scripted policies, active feature acquisition methods) are reported. This makes it hard to judge absolute task difficulty and whether current LLMs are close to clinically meaningful performance.

**Questions:**

1. Judge reliability. How often does the LLM‑as‑judge disagree with humans? Can you provide a human‑audited sample (e.g., 100 cases) with inter‑rater agreement for both accuracy and EQ?
2. Decoding settings. Which decoding hyperparameters were used for Table 1? Please reconcile the statements about "same decoding hyperparameters," "temperature=0," and Table 4’s per‑model defaults. If you did use per‑model defaults, can you rerun with a single shared decoding setting to test robustness?
3. Patient simulator size. Appendix B fixes the SP‑sim to Qwen2.5‑7B; Figure 4 varies patient model size (7B/14B/32B). Which configuration produced Table 1? If you ablated patient‑model size, please detail how the SP‑sim content was held constant across sizes.
4. Process metrics. Can you report turn distributions, % duplicate/unavailable test requests, stop‑too‑early vs stop‑too‑late rates, and a correlation between #turns and EQ/accuracy? This would substantiate the claimed failure modes.
5. Evidence verification strictness. EQ demands "verbatim support." How are synonyms/paraphrases handled in evidence (not diagnosis)? Could you provide sensitivity analyses with a looser lexical match?
6. Realism extensions. Do you plan to operationalize test costs/contraindications and expand the test catalog beyond what’s pre‑encoded in each case? How would you prevent label leakage if adding such dynamics?
7. Baselines & humans. Any plans to include human OSCE‑style baselines or active feature acquisition baselines to anchor performance relative to non‑LLM methods?

---

### Official Review · Reviewer_QowL · 2025-11-02

**Soundness:** 2
**Presentation:** 2
**Contribution:** 1
**Rating:** 2
**Confidence:** 5

**Summary:**

The authors present an evaluation method derived from existing datasets. They create two tasks: a zero-shot diagnosis task, where the complete vignette is provided to the LLM, and an active retrieval benchmark with the LLM playing the role of the patient and the other LLM being evaluated acting as the doctor. The method also measures the evidence provided by the LLM to support its diagnosis. They then generate a 468-case benchmark and assess a variety of models, demonstrating that models perform worse when presented with cases that require active retrieval of clinical information.

**Strengths:**

The inclusion of an evidence metric is novel and an interesting contribution. The number and variety of models tested provide a clear picture of the performance as well as evidence of scaling laws.

**Weaknesses:**

# Major

1) The reliance on previous benchmarks raises concerns about potential data contamination and outdated content. The lack of physician validation or guideline alignment also questions the validity (both internal and external) of this method.

2) The dataset is relatively small, and the coverage of key entities is likely incomplete. A detailed analysis of the coverage is necessary to interpret the results on the benchmark accurately.

3) I am not sure I understand the novelty (beyond evidence quality assessment) of the benchmark compared to previous simulators like AI Hospital [1].

4) While the authors correctly claim that there is a misalignment between evaluations and real-world clinical workflows, they do not discuss the limitations of their approach; for instance, requesting the history of present illness is not a trivial task. Prior work has demonstrated that simply navigating an EHR to retrieve relevant documents is a complex task involving multiple steps [2].

# Minor

1) The figures are difficult to read, especially the graphical abstracts (Fig. 1 and 2), which I only understood after reading the annex.

2) Bolding is overly used in my opinion and not discriminative enough (for example, I would use emph on model names and bold on task names to distinguish them).

3) Sections 3.2, 3.3, and 3.4 are overly complicated by mathematical notations that do not contribute much to the paper.

4) Some references are missing, for example, on L670.

# References

[1] AI Hospital: Benchmarking Large Language Models in a Multi-agent Medical Interaction Simulator (Fan et al., COLING 2025)

[2] MedAgentBench: A Virtual EHR Environment to Benchmark Medical LLM Agents (Jiang et al., NEJM AI 2025)

**Questions:**

# Major

1) I would like to see a physician validation or guideline cross-validation to ensure that the cases are accurate and up to date with clinical knowledge. For example, MedQA was released in 2019 while societies make new recommendations almost yearly (GOLD and GINA) [1-2].

2) MedMCQA is of poor quality overall; reliance on such datasets raises concerns about the quality of the generated data, especially without manual verification. Google abandoned MedMCQA since their MedGemini paper [3].

3) A coverage of medical topics would strengthen the validity of the dataset. For example, the authors could cross-reference topics in the benchmark with the topics found in the USMLE to ensure sufficient coverage of medical entities [4].

4) I would like to see a more complete discussion of the limitations of this approach and an explanation of the novelty of this method compared to prior work.

# References

[1] Global Initiative for Chronic Obstructive Lung Disease (GOLD; 2025)

[2] Global Initiative for Asthma (GINA; 2025)

[3] Capabilities of Gemini Models in Medicine (Saab et al., 2024)

[4] USMLE content outline (NBME, 2025)

---

### Official Review · Reviewer_kN9n · 2025-11-03

**Soundness:** 2
**Presentation:** 2
**Contribution:** 1
**Rating:** 2
**Confidence:** 5

**Summary:**

This paper argues that current medical benchmarks are flawed. They evaluate LLMs in a "full-context" setting, providing all patient information at once. This is unrealistic. Real clinical diagnosis requires "active evidence-seeking."

The authors introduce ROUNDS-Bench to address this. The benchmark splits evaluation into two tasks. Task 1 is the standard full-context (upper bound) evaluation. Task 2 is the novel "active evidence-seeking" task. In Task 2, a model only receives a chief complaint. It must then actively request information (HPI, PE, labs, imaging) from a "Standardized Patient Simulator" (SP-sim). The model must also decide when it has enough information to stop and make a diagnosis.

The core finding is that all LLMs (GPT-4o, Qwen, etc.) show a massive performance drop from Task 1 to Task 2. The paper also finds that "Evidence Quality" (citing correct, retrieved evidence) drops even more than diagnostic accuracy. This suggests models are not just bad at diagnosis under uncertainty, but also bad at the evidence-gathering process itself.

**Strengths:**

The paper's core premise is a strength. Evaluating active evidence-seeking is a clear and necessary next step for medical AI. The field is saturated with full-context benchmarks that overestimate real-world capability.

The Task 1 vs. Task 2 design cleanly isolates the full-context "reasoning" capability from the "active" process. This allows the authors to directly quantify the performance gap, which is a compelling contribution.

**Weaknesses:**

The paper significantly overlaps with MediQ (https://arxiv.org/abs/2406.00922) without proper citation. MediQ also designs an upper-bound task, benchmarks information-seeking, and provides a patient simulator. This paper's main addition seems to be the "evidence quality" metric, which relies on an unvalidated LLM-Judge (Qwen-32b) and lacks justification.

The related works section is weak. It focuses on static benchmarks but fails to discuss any state-of-the-art interactive diagnosis work (e.g., https://arxiv.org/pdf/2401.05654, https://arxiv.org/abs/2406.00922, https://arxiv.org/abs/2502.07143, https://arxiv.org/abs/2501.09484).

The parsing of patient records into predefined categories (Eq 1) is too idealistic. The paper provides no sanity check on how much of the source data can actually be parsed this cleanly.

While calling the active evidence-seeking scenario "realistic", the work presents a fairly unrealistic setting. The "Standardized Patient Simulator" (SP-sim) is not a patient simulator. It's a structured information oracle. The AI just requests a document like [Physical Examination], and the sim provides a block of text. This is not how real clinical inquiry works.

The main findings (e.g., the performance drop from full-context to active-seeking) have already been reported in MediQ, including analysis such as CoT performs better. This paper and its analysis do not appear to add significant new scientific insights.

**Questions:**

1. The SP-sim is a "gated oracle." How does this task test anything beyond a model's ability to follow a simple heuristic (e.g., always request HPI, then PE, then labs)? What prevents a simple, hard-coded script from scoring well on the process?

2. Why was Qwen2.5-32B-Instruct chosen as the sole evaluator? What is its agreement with human expert clinicians? Without this data, the paper's results are not verifiable.

3. The Evidence Quality metric requires exactly three items. Why three? This seems arbitrary. A sound diagnosis might rest on one critical test (a biopsy) or a collection of eight different findings. How does this rigid metric account for clinical reality?

4. How does the model request specific tests? The action space in (4) lists LAB(x) and IMG(x). How does the model know which 'x' to ask for? Is it guessing from a list, or generating free text (e.g., "Request Troponin")? This is a critical, unexplained detail.

5. The paper finds "think" models (with CoT) perform better. Is this simply because the Task 2 prompt (Fig 13) explicitly tells the model to output a rationale for its action? The experimental setup seems to be explicitly prompting for the very behavior it then "discovers" is better.

---

### Note · Authors · 2026-01-17

I have read and agree with the venue's withdrawal policy on behalf of myself and my co-authors.